# Exploring the Effectiveness of Isatin–Schiff Base as an Environmentally Friendly Corrosion Inhibitor for Mild Steel in Hydrochloric Acid

**Ahmed A. Al-Amiery** [1,2,*] 🆔, **Nadia Betti** [3], **Wan Nor Roslam Wan Isahak** [1] 🆔, **Waleed Khalid Al-Azzawi** [4] 🆔 **and Wan Mohd Norsani Wan Nik** [5] 🆔

1   Department of Chemical and Process Engineering, Faculty of Engineering and Built Environment, Universiti Kebangsaan Malaysia (UKM), Bangi 43000, Malaysia
2   Energy and Renewable Energies Technology Center, University of Technology-Iraq, Baghdad 10001, Iraq
3   Materials Engineering Department, University of Technology-Iraq, Baghdad 10001, Iraq
4   Medical Technical College, Al-Farahidi University, Baghdad 10001, Iraq
5   Faculty of Ocean Engineering Technology and Informatics, Universiti Malaysia Terengganu, Kuala Terengganu 20300, Malaysia
*   Correspondence: dr.ahmed1975@gmail.com or dr.ahmed1975@ukm.edu.my

**Abstract:** A recent study has shown that Schiff base OHMHI is an effective inhibitor of the corrosion of mild steel in acidic media. The study utilized weight loss measurements and electrochemical techniques, such as EIS and potentiodynamic polarization, to analyze the corrosion inhibition efficiency of OHMHI. The results of the study show that the presence of OHMHI in the corrosive environment significantly reduced the corrosion rate of mild steel and increased its corrosion resistance. The impedance spectra analysis indicated that OHMHI was adsorbed on the surface of mild steel, providing a protective layer. The potentiodynamic polarization study confirmed the protective role of OHMHI by showing an increase in the passive current density of the mild steel in the presence of OHMHI. The inhibitory efficiency of OHMHI was found to be 96.1%, indicating that it is an effective corrosion inhibitor for mild steel. The study also investigated the optimal conditions for the use of OHMHI as a corrosion inhibitor, with a concentration of 0.5 mM and a temperature of 303 K being chosen. The Langmuir adsorption isotherm concept was used to demonstrate the physical and chemical adsorption of OHMHI on the surface of mild steel. Morphological investigations of the uninhibited and inhibited surfaces of the mild steel specimen were examined using scanning electron microscopy (SEM) analysis. Furthermore, computational investigations using density functional theory (DFT) and experimental data were merged to explore the corrosion inhibition efficiency and mechanism of inhibition. Although the results are promising, further studies are needed to determine the long-term effects of OHMHI on mild steel corrosion and to evaluate its effectiveness under different environmental conditions. Overall, the study highlights the potential of OHMHI as an effective corrosion inhibitor for mild steel in acidic media.

**Keywords:** indolin; corrosion inhibitor; mild steel; weight loss; DFT

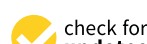



## 1. Introduction

Mild steel, as a low-carbon steel, is susceptible to corrosion in corrosive solutions such as hydrochloric acid. To reduce corrosion, organic inhibitors can be added to the solution [1–3]. These inhibitors work by adsorbing onto the steel surface and forming a protective layer, reducing the rate of reaction between the steel and the corrosive solution. Some common organic inhibitors include benzotriazoles, benzimidazoles, and organic phosphates [4,5]. It is important to note that the choice of inhibitor and its concentration should be optimized based on the specific application and the concentration of the corrosive solution [6]. Organic inhibitors for mild steel include lignin, tannins, polyphenols, fatty

acids, proteins, alkaloids, acetates, phosphates, sulfides, and organic acids such as formic, acetic, citric, and oxalic acids [7,8]. Due to the bonding of lone pairs and/or electrons with the metal surface, organic molecules of this kind are easily adsorbed on the mild steel surface [9]. This type of adsorption is known as physisorption, which occurs through van der Waals forces. The adsorption of organic molecules on mild steel surfaces can alter the surface properties of the metal, leading to changes in its chemical behavior and reactivity. Additionally, the presence of organic molecules on the surface of mild steel can also have a significant impact on its corrosion resistance [10,11]. The adsorption of organic molecules on mild steel surfaces can also influence the growth of microorganisms on the metal surface, which is of particular importance in the biomedical and food industries [12–14]. The adsorption of organic molecules can prevent the growth of bacteria and fungi, leading to an increase in the shelf life and preservation of food products [15]. In conclusion, the adsorption of organic molecules on mild steel surfaces is an important phenomenon that plays a crucial role in various industrial processes. Further research and development in this field may lead to improved processes and applications, such as the creation of functional coatings and surfaces.

Scientists have recently developed a significant interest in Schiff bases (NCH), which are effective corrosion inhibitors and can prevent metal corrosion in various applications. These compounds act as a protective barrier on the metal surface, preventing the formation of rust and other corrosion products. Schiff bases are used in a wide range of industries, including oil and gas, marine, and construction, to protect metals and prevent corrosion [16–18]. The mechanism of action of Schiff bases involves the formation of a thin film on the metal surface that is resistant to corrosive agents such as water and oxygen. This film is stable and does not easily break down, ensuring long-term protection of the metal. Additionally, Schiff bases also act as cathodic inhibitors, meaning that they reduce the rate at which electrons are lost from the metal surface, thus slowing down the corrosion process [19].

One of the advantages of using Schiff bases is their compatibility with a wide range of metals, including steel, aluminum, and other alloys. This makes them an ideal choice for industries that require protection of multiple metal surfaces. Moreover, Schiff bases are environmentally friendly, with low toxicity levels and minimal impact on the ecosystem [20]. In conclusion, the use of Schiff bases is an effective and environmentally friendly solution for preventing metal corrosion. Their ability to protect a wide range of metals and their stability over time make them an attractive option for industries that require long-term corrosion protection [21]. In comparison with inhibitors containing only one of the two elements, those that combine both nitrogen and sulfur are incredibly effective at preventing corrosion. The manuscript titled "Isatin Schiff base as Green Corrosion Inhibitor for Mild Steel in Hydrochloric Acid Environment" discusses the effectiveness of Isatin Schiff base in preventing the corrosion of mild steel in a hydrochloric acid environment. The study compares the performance of inhibitors containing only one of the two elements (nitrogen or sulfur) with those containing a combination of both. One example of an inhibitor that contains nitrogen is pyridine, which has been shown to be effective in preventing the corrosion of mild steel in hydrochloric acid environments. Similarly, thiourea is an example of an inhibitor that contains sulfur and that has been found to be effective in preventing corrosion [22]. However, the study found that inhibitors containing both nitrogen and sulfur were significantly more effective than those containing only one of the two elements. For example, an Isatin Schiff base containing both nitrogen and sulfur was found to have excellent corrosion inhibition performance [23]. The presence of sulfur in the molecule is known to enhance the corrosion inhibition performance, but the absence of sulfur does not necessarily diminish the effectiveness of the inhibitor. The study highlights the importance of considering the combination of heteroatoms in designing effective corrosion inhibitors. However, one family of compounds that includes both nitrogen and sulfur is heterocyclic rings, whose Schiff bases have been documented, and whose effects on corrosion inhibition have been studied extensively [24]. These heterocyclic compounds exhibit excellent

inhibition efficacy against a variety of corrosive environments, including acidic, alkaline, and neutral conditions. The presence of both nitrogen and sulfur in the same molecule enhances the overall performance of the inhibitor, as these elements are known to form strong complexes with metal ions and reduce the corrosion rate [25]. The mechanism of inhibition is believed to involve adsorption of the inhibitor molecules on the metal surface, forming a protective film that blocks the transfer of electrons and slows down the corrosion reaction. The heterocyclic ring structure of these compounds also contributes to their effectiveness, as it enhances the stability of the adsorbed layer and prevents the desorption of the inhibitor molecules [26]. In addition, these compounds show low toxicity and environmental impact, making them an attractive option for industrial applications. They can be used in a variety of systems, including oil and gas pipelines, water systems, and electrical components, to prevent corrosion and extend the lifespan of these assets [27]. In conclusion, combination inhibitors containing both nitrogen and sulfur, specifically heterocyclic compounds, have proven to be highly effective in preventing corrosion in a variety of environments. They offer a safe and environmentally friendly alternative to traditional corrosion inhibitors, making them a valuable tool in the fight against corrosion [28]. The aim of this study was to use OHMHI (Figure 1) as a corrosion inhibitor for mild steel in hydrochloric acid solution. The study was conducted to determine the effectiveness of OHMHI as a corrosion inhibitor for mild steel in hydrochloric acid solution and involved measuring the corrosion rate and inhibition efficiency and analyzing the surface of the mild steel samples treated with different concentrations of OHMHI. This research article presents an innovative approach that uses Isatin Schiff base (OHMHI) as a green corrosion inhibitor for mild steel in a hydrochloric acid environment. The study investigates the inhibitory efficiency of OHMHI using weight loss measurements, electrochemical impedance spectroscopy (EIS), and potentiodynamic polarization techniques. Additionally, the study explores the optimal experimental conditions, including the concentration of OHMHI and temperature.

**Figure 1.** The chemical structure of the tested inhibitor.

Furthermore, the paper examines the physical and chemical adsorption of OHMHI on the surface of mild steel using the Langmuir adsorption isotherm concept. Moreover, scanning electron microscopy (SEM) analysis was used to investigate the morphological investigations of the mild steel specimen's uninhibited and inhibited surfaces. Finally, the study uses both computational investigations using density functional theory (DFT) and experimental data to explore the corrosion inhibition efficiency and mechanism of inhibition. Overall, this paper's innovation lies in the using of OHMHI as a green corrosion inhibitor for mild steel and the comprehensive investigation of its inhibitory efficiency and its mechanism of inhibition using various experimental and computational techniques.

## 2. Social and Economic Effects of Corrosion

Corrosion is a natural process that occurs when metals and alloys react with their environment, leading to a gradual deterioration of their physical and chemical properties. The effects of corrosion can be significant, both in terms of social and economic impacts. From a social perspective, corrosion can have a range of negative effects on human health and safety. For example, corrosion of metal structures such as bridges, pipelines, and buildings can weaken their structural integrity and increase the risk of collapse or failure. Corrosion can also lead to the release of hazardous materials, such as lead, into the environment, posing a threat to human health. Economically, corrosion can have a significant impact on industries and infrastructure. The direct costs of corrosion include the repair

or replacement of damaged equipment, lost productivity due to downtime, and increased maintenance and inspection costs. According to a report by NACE International, the global cost of corrosion was estimated to be $2.5 trillion in 2019, representing approximately 3.4% of global GDP [29]. In addition to these direct costs, corrosion can also have indirect economic impacts, such as the reduced competitiveness of businesses, decreased property values, and reduced tax revenues for governments. For example, the corrosion of water supply infrastructure can lead to reduced water quality and increased health risks, which can negatively impact tourism and local economies [30]. To mitigate the negative effects of corrosion, it is essential to implement corrosion prevention and control measures. These measures can include the use of corrosion-resistant materials, protective coatings, and cathodic protection systems. Implementing such measures can help to reduce the direct and indirect costs of corrosion and improve human health and safety. In conclusion, corrosion can have significant social and economic impacts, including risks to human health and safety, direct costs associated with equipment repair and replacement, and indirect costs such as reduced competitiveness and decreased property values. To mitigate these impacts, it is essential to implement effective corrosion prevention and control measures.

## 3. Materials and Methods

### 3.1. Materials

Mild steel coupons, from Metal Samples Company, each with a composition (wt. %) of carbon (0.210), manganese (0.050), silicon (0.380), aluminum (0.010), sulphur (0.050), phosphorus (0.090), and a remainder of iron, were subjected to weight loss measurements and electrochemical techniques to study their corrosion behavior. The coupon dimensions for weight loss measurements were 4.0 cm × 2.5 cm × 0.1 cm, and for electrochemical techniques, they were 1.0 cm × 1.0 cm × 0.1 cm. In all tests, the mild steel coupons were pre-treated using a sequence of steps. First, the samples were ground with emery paper of grades 400, 600, and 1200. Then, they were thoroughly washed with double-distilled water to remove any residual particles. Next, the samples were degreased using ethanol and, finally, were dried at room temperature before use [31].

### 3.2. Test Solutions

Hydrochloric acid (1 M) was prepared by diluting 37% analytical grade HCl (Merck-Malaysia) with double-distilled water. The concentration of OHMHI was adjusted by adding varying amounts of the compound to the 1 M HCl solution. The solution was thoroughly mixed to ensure a homogeneous solution was obtained. The corrosive properties of the solution were tested by immersing metal samples into the solution for a specified period of time. The metal samples were then removed and evaluated for any signs of corrosion or degradation. This was achieved by visually inspecting the samples for changes in appearance, such as discoloration, pitting, or loss of thickness, and by measuring the change in weight of the samples to determine the extent of corrosion. The results of these tests were used to determine the efficacy of OHMHI as a corrosion inhibitor in acidic media.

### 3.3. Gravimetric Analysis

Corrosion rate ($C_R$), inhibition efficiency ($IE\%$) and surface coverage ($\theta$) were acquired by gravimetric method as per the national association of corrosion engineer's standard (NACE) [31,32]. Mild steel coupons were immersed in 1 M HCl for 5 h in the absence and presence of OHMHI as a corrosion inhibitor at various concentrations (0.1, 0.2, 0.3, 0.4 and 0.5 mM) and at different temperatures (303–333 K). In order to study the effect of time, mild steel coupons were exposed to corrosive solution at 303 K, for various immersion times (1, 5, 10, 24, and 48 h), in the presence of OHMHI as corrosion inhibitor at various concentrations (0.1, 0.2, 0.3, 0.4 and 0.5 mM). The corrosion rate, inhibition efficiency and surface coverage as corrosion parameters were determined according to Equations (1)–(3) [33]:

$$C_R = \frac{W}{at} \tag{1}$$

$$IE\% = \left[1 - \frac{C_{R(i)}}{C_{Ro}}\right] \times 100 \tag{2}$$

$$\theta = 1 - \frac{C_{R(i)}}{C_{Ro}} \tag{3}$$

### 3.4. Adsorption Isotherms

One can gain important insights into the characteristics of the tested compounds by analyzing the adsorption isotherm type. The degree of surface coverage ($\theta$) for inhibitors can be determined by applying different adsorption isotherms, such as Langmuir, Frum-kin, and Temkin. In this research, weight loss measurements were utilized to determine the values of surface coverage ($\theta$) for various inhibitor concentrations in acidic media. The weight loss measurements were carried out using a scale with a sensitivity of 0.001 g. The samples used in the experiments had an area of 1.0 cm $\times$ 1.0 cm $\times$ 0.1 cm.

### 3.5. Electrochemical Method

Electrochemical assessment was conducted using the Gamry Instrument Potentio-stat/Galvanostat/ZRA type REF 600 and the DC105 and EIS300 software developed by Gamry (Warminster, PA, USA). The testing was carried out in a three-electrode setup, where the reference electrode was a saturated calomel electrode, and the working electrode was made of metal. The working electrode had an exposed area of 4.5 cm$^2$ and was thoroughly cleaned according to ASTM G1-03 [31]. (SCE). To obtain the average, each test was undertaken three times.

Equation (4) illustrates how the inhibitive efficacy was evaluated:

$$IE(\%) = \frac{i_{corr} - i_{corr(inh)}}{i_{corr}} \times 100 \tag{4}$$

where $i_{corr(inh)}$ and $i_{corr}$ represent the current density in the presence/absence of the tested inhibitor molecules, respectively.

In order to maintain a constant voltage of 303 K, the measurements were taken 30 min after immersing the working electrode in an acidic solution. Tafel curves were generated by scanning from 0.25 V to +0.25 V SCE at a rate of 0.5 mV/s. The electrochemical impedance tests (EIS) results were analyzed using the appropriate equivalent circuits with the assistance of the Gamry Echem Analyst tool.

Formula (5) was used to compute the inhibitory activity (*IE%*) from the charge transfer impedance:

$$IE(\%) = \frac{R'_{ct} - R_{ct}}{R'_{ct}} \times 100 \tag{5}$$

where $R'_{ct}$ and $R_{ct}$ are the charge transfer resistances in the presence and absence of the tested inhibitor molecules.

### 3.6. SEM

The study utilized mild steel specimens that measured 4.0 cm $\times$ 2.5 cm $\times$ 0.1 cm. To prepare the samples, a range of emery paper was used to abrade the surface. Subsequently, they were thoroughly cleaned with distilled water and acetone. The specimens were then submerged in uninhibited and inhibited acidic solutions, specifically 1 M HCl, at the optimal inhibitor concentration of 0.5 mm. The immersion process was conducted at 303 K for a period of 5 h. Afterward, the mild steel coupons were washed with distilled water, dried using an oven, and subjected to scanning electron microscopy. Imaging was conducted using a Model Zeiss MERLIN Compact FESEM, which was facilitated by the UKM electron microscopy unit.

*3.7. Computational Investigations*

The quantum chemistry calculations were carried out using Gaussian 09 software [34]. The inhibitor structure was optimized in the gas phase using the B3LYP method and a 6-31G++(d,p) basis set. The ionization potential ($I$) and electron affinity ($A$) were determined as $E\_HOMO$ and $E\_LUMO$, respectively, based on Koopman's theorem [35,36]. To calculate ($I$) and ($A$), Equations (6) and (7) were employed.

$$I = -E_{HOMO} \tag{6}$$

$$A = -E_{LOMO} \tag{7}$$

The determination of quantum chemical parameters, including electronegativity ($\chi$), hardness ($\eta$), softness ($\sigma$), and transferred electrons fractional number ($\Delta N$) [37,38], was conducted via the Equations (8)–(11):

$$\chi = \frac{I + A}{2} \tag{8}$$

$$\eta = \frac{I - A}{2} \tag{9}$$

$$\sigma = \eta^{-1} \tag{10}$$

$$\Delta N = \frac{7 - \chi_{inh}}{2(\eta_{inh})} \tag{11}$$

Here, $\chi_{inh}$ and $\eta_{inh}$ are the electronegativity and the hardness of inhibitor, respectively (Note: $\chi_{Fe}$ = 7 eV, $\eta_{Fe}$ = 0 eV).

## 4. Results and Discussion

*4.1. Gravimetrical Measurements*

4.1.1. Effect of Concentration

As the concentration of OHMHI increased, the anticorrosion performance improved. At a temperature of 303 K and a five-hour exposure period, the corrosion rate decreased with increased OHMHI concentration. The highest inhibitory efficiency (95.8%) was achieved at a concentration of 0.5 mM OHMHI (Figure 2). An aromatic group increases the electron density at the active sites, enhancing the interactions of OHMHI molecules with iron atoms located on the metallic substrate, thus enhancing the protective properties of OHMHI. The presence of amide and hydroxyl groups in the OHMHI structure also contributes to its inhibition performance, as they form coordination bonds with the metal surface. These coordination bonds increase the adsorption of OHMHI on the metal surface, resulting in a more efficient protective layer against corrosion. The effect of different concentrations of corrosion inhibitors on mild steel in 1 M HCl solution will vary. Generally, higher concentrations of corrosion inhibitors will result in better protection against corrosion, but the optimal concentration may depend on the specific inhibitor and on other variables such as temperature and the presence of other ions in the solution. Additionally, some inhibitors may become less effective at high concentrations or even become corrosive themselves. In the weight loss test for metallic coupons in uninhibited and inhibited corrosive solutions, it was found that the presence of OHMHI (a corrosion inhibitor) protected the metallic surface from the corrosive solution [39–46].

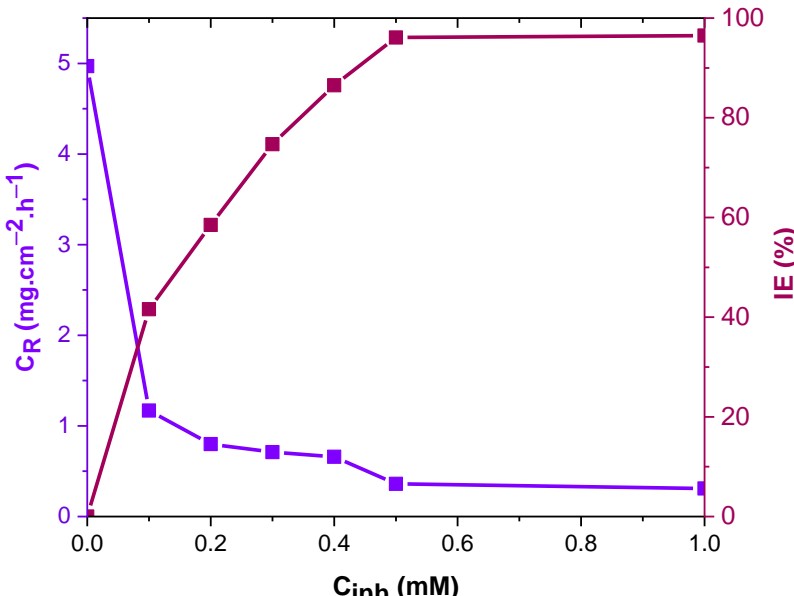

**Figure 2.** The corrosion rate and inhibition efficiency of mild steel exposed to a 1 M HCl solution for 5 h at 303 K were studied in relation to different concentrations of OHMHI.

Overall, the molecular structure and chemical composition of OHMHI play a crucial role in its ability to prevent corrosion. The presence of heterogeneous atoms and functional groups, such as aromatic groups, contributes to the strong interaction of OHMHI with the metal surface, resulting in an effective corrosion inhibitor [47]. OHMHI molecules, as corrosion inhibitors, have efficient characteristics for reducing/inhibiting acidic corrosion by increasing the concentration of OHMHI up to 0.5 mM. This is due to the adsorption of these molecules onto the surface of the tested coupons, producing a protective film covering the mild steel surface. When the inhibitor concentration rises above 0.5 mM and reaches 1.0 mM, the OHMHI molecules that have been adsorbed on the surface of the substrate allow the inhibitory efficiency to remain practically constant. This is because the protective film formed by the OHMHI molecules acts as a barrier to the corrosive agents and prevents the substrate from being attacked. The protective film is also capable of absorbing the corrosive species and neutralizing them, further reducing the rate of corrosion. The presence of OHMHI molecules in the corrosive medium enhances the stability of the protective film and slows down the corrosion rate. This is due to the molecule's ability to form strong bonds with the metal surface, creating a dense and stable layer that acts as a barrier to corrosive agents.

There are various studies that have investigated the effect of different concentrations of corrosion inhibitors on mild steel corrosion rates in acid solutions. For instance, a study by Fang [48] examined the performance of several organic corrosion inhibitors in 1 M HCl solution, including imidazole, benzotriazole, and 2-mercaptobenzothiazole. The researchers found that higher inhibitor concentrations generally led to lower corrosion rates, but that the optimal concentration varied depending on the inhibitor type. The impact of natural inhibitors on mild steel corrosion in 1 M HCl solution has been investigated in several studies. For example, Mousa and Al-Mobarak [49] examined the influence of various concentrations of pomegranate peel extract on corrosion rates. They observed that the inhibitor concentration had a direct effect on corrosion rate reduction, but that inhibition efficiency peaked at a specific concentration.

Similarly, Olasunkanmi [50] studied the impact of castor oil as a green inhibitor on mild steel corrosion in the same solution. They found that corrosion rates decreased with increasing inhibitor concentration up to a certain point, after which inhibition efficiency declined. These results suggest that the optimal inhibitor concentration is reliant on a variety of factors, including the type of inhibitor, its concentration, and the solution's

conditions. Consequently, it is recommended to consult literature specific to the chosen inhibitor to determine the ideal concentration for corrosion inhibition. In conclusion, OHMHI molecules have proven to be effective corrosion inhibitors in acidic environments. They are able to reduce the corrosion rate and protect the metal surface by forming a stable and protective film. The optimal concentration of OHMHI to achieve the best inhibitory efficiency is 0.5 mM, and increasing the concentration beyond this point has little effect on inhibitory efficiency.

### 4.1.2. Effect of Immersion Periods

In order to determine the effect of immersion periods on the effectiveness of OHMHI's ability to reduce/prevent corrosion, mild steel was exposed to 1 M HCl supplemented with different concentrations of OHMHI (0.1 to 1.0 mM) for different immersion times (1 h to 48 h) at 303 K (Figure 3).

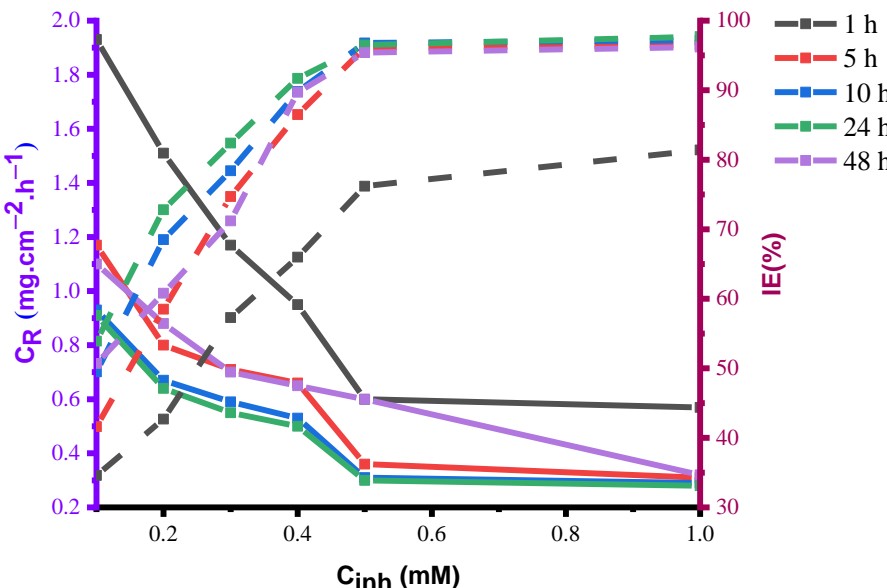

**Figure 3.** OHMHI concentrations effect on the rate of corrosion and inhibitory efficiency of metallic coupons in 1 M HCl for various immersion periods at 303 K.

The results show that, as the concentration of OHMHI increased, the rate of corrosion decreased significantly. At 0.1 mM OHMHI, the corrosion rate was found to be very high, but as the concentration increased to 0.5 mM, the corrosion rate decreased by almost 50%. At 1.0 mM OHMHI, the corrosion rate was found to be the lowest, with a decrease of almost 70% compared with the 0.1 mM OHMHI concentration.

The immersion time also had a significant effect on the effectiveness of OHMHI in reducing corrosion. The longer the immersion time, the more effective OHMHI was in reducing corrosion. For example, at 1.0 mM OHMHI, the corrosion rate was found to be the lowest after 48 h of immersion, with a decrease of 80% compared with the 1-hour immersion time.

In conclusion, the effect of immersion periods on the effectiveness of OHMHI in reducing corrosion was found to be significant. The results show that a higher concentration of OHMHI and a longer immersion time were effective in reducing corrosion. This implies that OHMHI could be a potential alternative to traditional corrosion inhibitors for the protection of metal surfaces.

The inhibitory efficiency of OHMHI molecules on metal substrates shows a rapid improvement with an increasing immersion period of up to 10 h. After this period, it remains approximately constant up to 24 h, and then gradually decreases from 24 to 48 h. The amount of OHMHI molecules adsorbed onto the metal substrate increases with immersion duration, which enhances inhibitory efficiency due to the increased concentration of

OHMHI molecules. The metal substrate acts as a catalyst for the chemical reaction between the OHMHI molecules and the metal surface, resulting in a uniform coating of OHMHI molecules that forms a barrier to inhibit corrosion.

However, if the immersion period exceeds 24 h, the concentration of OHMHI molecules decreases due to depletion caused by the reaction with the metal surface, resulting in a gradual decrease in inhibitory efficiency. The inhibitory efficacy is influenced by other factors such as temperature, pH, and OHMHI solution concentration, which should be carefully controlled for optimal inhibitory efficiency. The inhibitory efficacy of OHMHI molecules on metal substrates is a complex process that depends on several factors, including immersion period, temperature, pH, and concentration. The maximum inhibitory efficacy is achieved between 10 to 24 h of immersion period. Moreover, the strong hydrogen bonding and coordination interactions between the inhibitor molecules and the metal substrate contribute to the stability and efficacy of OHMHI as a corrosion inhibitor in acidic environments. Additionally, the high adsorption density of OHMHI leads to physisorption (van der Waals forces, hydrogen bonding, or hydrophobic interactions) and chemisorption (coordination interactions to form between inhibitor molecules and iron atoms on the mild steel substrate). If some inhibitor molecules escape the surface, the effective area that the inhibitor covers may be reduced, and the inhibitory activity may be decreased. Nonetheless, the OHMHI layer's stability is high, proven by the reasonably high inhibitory efficacy during the prolonged exposure period, which blocks the access of corrosive agents to the metal surface, reduces the rate of corrosion, and thus increases the inhibitory efficacy of OHMHI.

### 4.1.3. Effect of Temperature

The corrosion inhibition of metallic substrate in inhibited corrosive solution with different concentrations of OHMHI (0.1–1.0 mM) was studied using the mass reduction technique after 5 h of immersion at various temperatures (303–333 K). At the same OHMHI concentration, the rate of corrosion increased, and corrosion efficiency declined as temperature increased from 303 to 333 K (Figure 4). This suggests that the effect of temperature on the inhibition efficiency of OHMHI is significant. A higher temperature may lead to an increase in the activity of corrosive species in the solution, thus reducing the effectiveness of OHMHI as a corrosion inhibitor. The decrease in inhibition efficiency at higher temperatures can also be attributed to the thermodynamic instability of OHMHI at elevated temperatures, causing it to break down and become less effective in inhibiting corrosion [51].

Additionally, the increase in temperature may also increase the rate of reaction between OHMHI and the metallic substrate, leading to the consumption of the inhibitor, which further reduces its inhibition efficiency. In conclusion, temperature plays a crucial role in the efficacy of OHMHI as a corrosion inhibitor. The inhibition efficiency of OHMHI decreases as temperature increases, highlighting the importance of considering temperature when evaluating the performance of corrosion inhibitors.

OHMHI performed best at the temperature of 303 K. A decrease in inhibitory potency with rising temperature at all doses points to physisorption (physical adsorption). OHMHI molecules are also lost from the metallic surface of at high temperatures due to desorption [52]. Inhibition efficiencies of OHMHI have been investigated at various temperatures of 303, 313, 323, and 333 K. OHMHI shows significant inhibition efficiency at 0.5 mM with considerable inhibitive performance: 96.1% at 303 K, 95.4% at 313 K, 93.1% at 323 K, and 85.5% at 333 K; whereas at 1.0 mM the inhibition efficiencies were 96.5% at 303 K, 96.2% at 313 K, 94.7% at 323 K, and 93.9% at 333 K (Figure 4). Figure 4 shows that, as the temperature increases, the inhibition efficiency of OHMHI decreases slightly, especially at the highest tested inhibitor concentration (1 mM). These findings can be explained by referring to the chemisorption and physisorption mechanisms. Furthermore, the interactions between OHMHI molecules and the d-orbitals of iron atoms located on the mild steel surface form coordination bonds through the transfer of ion pairs of heteroatoms from the inhibitor

molecules to Fe-orbitals (chemical adsorption). Additionally, the interactions between the mild steel surface and inhibitor molecules were based on physical adsorption mechanisms, namely van der Waals forces. These findings suggest that the inhibition efficiency of OHMHI is primarily dependent on the temperature of the environment. The decrease in inhibitive performance at higher temperatures can be attributed to both chemical and physical adsorption mechanisms. The formation of coordination bonds between the OHMHI molecules and the iron atoms on the mild steel surface through the transfer of ion pairs is an example of chemical adsorption. On the other hand, the interactions between the mild steel surface and OHMHI molecules due to van der Waals forces are an example of physical adsorption. These findings can be used to optimize the conditions under which OHMHI can be used as a corrosion inhibitor for mild steel.

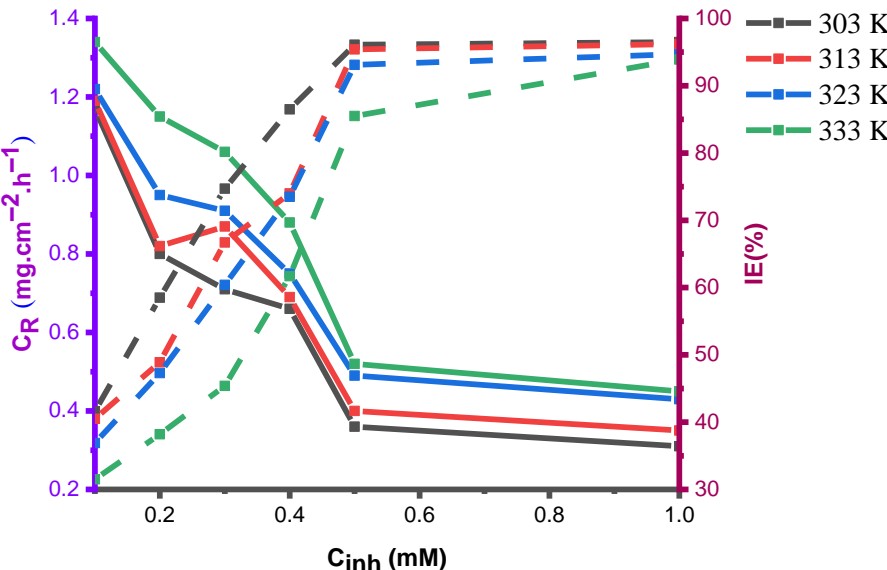

**Figure 4.** OHMHI concentrations effect on the rate of corrosion and inhibitory efficiency of metallic coupons in 1 M HCl at different temperatures for 5 h.

In order to evaluate the corrosion process activation parameters, Arrhenius Equation (12) has been utilized and is exhibited in Figure 5:

$$log C_R = log K - (E_a / 2.303RT) \qquad (12)$$

It is believed that the metal substrate dissolves slowly due to inhibiting processes that require a higher activation energy compared with uninhibited processes. Furthermore, the activation energy rises as the inhibitor concentration increases. This suggests that, as the concentration increases, the inhibitor acts as a more formidable physical barrier against the corrosion process, leading to a decreased corrosion rate. This occurs because the electrons needed for the dissolution reaction must surmount a higher activation energy in the presence of the inhibitor. The inhibitor creates a protective layer on the metal surface, preventing corrosive agents from accessing the metal. As the inhibitor concentration grows, the protective layer becomes thicker and more durable, further decreasing the corrosion rate.

Moreover, the inhibitor can also adsorb on the metal surface and create a strong chemical bond that prevents the corrosion reactions from proceeding. This bond provides an additional barrier to the corrosion process and reduces the corrosion rate even further.

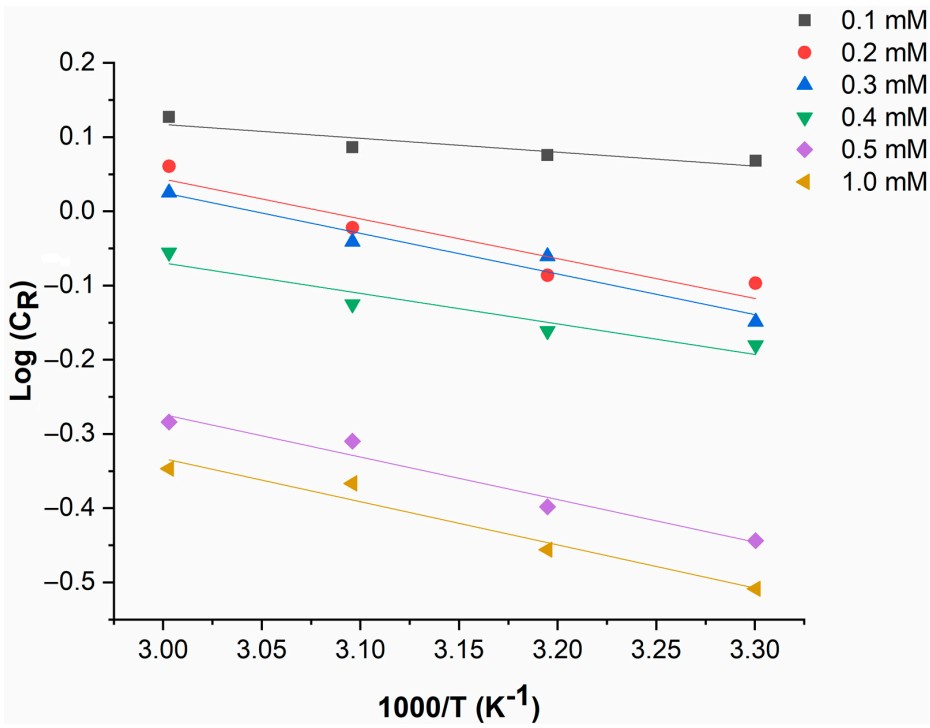

**Figure 5.** Arrhenius plot of logarithm corrosion rate versus 1000/temperature for various inhibitor concentrations.

In conclusion, the use of inhibitors in metal corrosion is an effective method of slowing down the corrosion process. By increasing the activation energy required for the corrosion reactions and forming a protective layer or chemical bond on the metal surface, inhibitors can significantly reduce the rate of corrosion and prolong the lifetime of metal structures. As the temperature rises, fewer inhibitor molecules are able to bind to the metal substrate, increasing the corrosion rate. Based on Equation (13), and Figure 6, (which represent the relation between log $C_R$/T and 1000/T for the immersed mild steel in corrosive solution), we can determine thermodynamic parameters such as the activation-enthalpy (ΔHa) and activation-entropy (ΔSa):

$$log\left\{\frac{C_R}{T}\right\} = \left\{log\left[\frac{R}{Nh}\right] + \left[\frac{\Delta S_a}{2.303R}\right]\right\} - \left[\frac{\Delta H_a}{2.303RT}\right] \tag{13}$$

where $N$ and $h$ are the number of Avogadro, and the Planck constant, respectively.

The values of ΔHa and ΔSa were determined based on the slope and intercept of the plot in Figure 6, and these values are shown in Table 1. The exothermic nature of the metallic substrate immersion process is reflected by the positive values of ΔH. This means that energy must be supplied to the system for the reaction to occur, which indicates an increase in bond energy and an increase in the stability of the substrate. The positive values of ΔHa can be attributed to the formation of a new oxide layer on the substrate surface, which requires energy input to break the existing bonds and form new bonds. On the other hand, the negative values of ΔSa indicate that the reaction is exothermic in nature and releases heat to the environment. This can be interpreted as a decrease in the entropy of the system due to the formation of a new, more ordered oxide layer on the substrate surface. In summary, the values of ΔHa and ΔSa provide valuable information about the thermodynamic properties of the reaction between the metallic substrate and the electrolyte solution. They can be used to predict the stability and energy efficiency of the substrate during the anodization process. The corrosion rate with a low value is due to the activation kinetic parameters, which can be referred to as the increase in the activation enthalpy with the increase in the concentration of the tested inhibitor. Based on

Table [1], we can see that the $\Delta Ha$ values for the uninhibited and inhibited solutions were 60.43 kJ·mol$^{-1}$ and 42.54 kJ·mol$^{-1}$, respectively (at the optimum inhibitor concentration), whereas the values of $\Delta Sa$ in the uninhibited and inhibited solutions were 57.58 J/(mol K) and 138.92 J/(mol K), respectively (at the optimum inhibitor concentration). The $\Delta H$ and $\Delta S$ values were evaluated from the slope and intercept, respectively, of the plot in Figure [6]. The metallic substrate immersion corrosion process is exothermic according to the positive enthalpy values of activation in uninhibited and inhibited solutions.

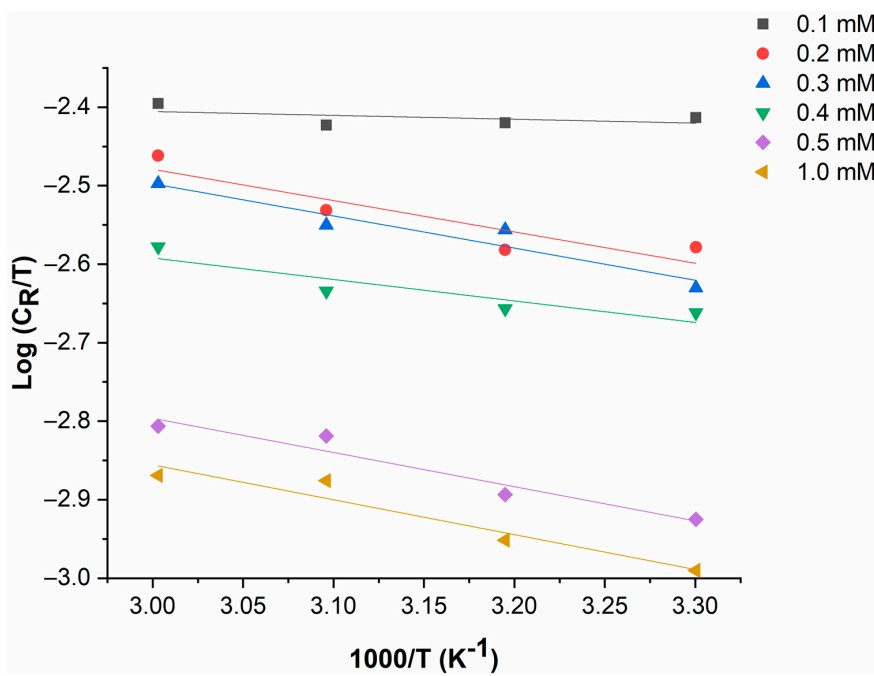

**Figure 6.** The relation between log $C_R/T$ and 1000/T for the immersed mild steel in corrosive solution with various inhibitor concentrations.

**Table 1.** The values of thermodynamics parameters for the immersed mild steel in 1 M HCl with various inhibitor concentrations.

| $C$ (mM) | $E_a \left( \text{kJ·mol}^{-1} \right)$ | $\Delta H_a \left( \text{kJ·mol}^{-1} \right)$ | $\Delta S_a \left( \text{J·mol}^{-1} \text{ K}^{-1} \right)$ |
|---|---|---|---|
| 0.0 | 63.11 | 60.43 | 57.58 |
| 0.1 | 55.36 | 53.73 | 117.09 |
| 0.2 | 50.66 | 51.79 | 120.06 |
| 0.3 | 47.04 | 48.37 | 123.79 |
| 0.4 | 42.81 | 45.64 | 129.47 |
| 0.5 | 40.47 | 42.54 | 138.92 |
| 1.0 | 38.25 | 40.46 | 147.76 |

The decrease in the activation energy ($\Delta H_a$) value in an inhibited solution compared with an uninhibited solution indicates that the presence of the inhibitor has reduced the energy barrier for the corrosion process to occur. This means that the inhibitor has made corrosion thermodynamically more favorable. However, the decrease in $\Delta H_a$ also means that the inhibitor has decreased the rate of corrosion by slowing down the reaction.

In addition, the increase in the entropy change ($\Delta S_a$) value in the inhibited solution compared with the uninhibited solution indicates that the inhibitor has increased the disorderliness of the corrosion process. This increase in entropy is also an indication of a decreased corrosion rate. However, it is important to note that, while the decrease in $\Delta H_a$ favors corrosion, the increase in $\Delta S_a$ works against it. Ultimately, the net effect of the inhibitor on corrosion depends on the balance between these two factors.

In conclusion, the activation energy and entropy results show that the tested inhibitor has a positive effect on the decrease of the corrosion rate of the metallic substrate. The results suggest that the inhibitor reduces the activation energy required for the corrosion process to occur, and increases the entropy of the process, both of which lead to a lower corrosion rate.

### 4.1.4. Adsorption Isotherm

The adsorption isotherm facilitates the comprehension of the interaction mechanism between the particles of the tested inhibitor and the metallic substrate. In order to determine which isotherms would correctly match the findings, surface coverage ($\theta$) values for the OHMHI that were gathered through gravimetric tests were used. Temkin, Freundlich, and Langmuir isotherms were employed to examine the adsorption mechanism. The results show that the adsorption of the metal ions onto the adsorbent was best described by the Langmuir isotherm model, indicating that the adsorption process was monolayer and the metal ions were adsorbed onto the surface of the adsorbent with a fixed maximum capacity. The Temkin isotherm model indicated that the adsorption process was controlled by the heat of adsorption, suggesting that the metal ions adsorbed onto the surface of the adsorbent through physisorption. On the other hand, the Freundlich isotherm model suggested that the adsorption process was controlled by the heterogeneity of the adsorbent surface and that the adsorption was a multi-layer process. These results indicate that the adsorption of metal ions onto the adsorbent was a combination of physisorption and chemisorption, with the former being the dominant mechanism. The Langmuir isotherm model provides a useful tool for determining the maximum capacity of the adsorbent and the metal ion uptake under different conditions. In addition, the Temkin and Freundlich isotherms give insight into the nature of the adsorption process and the role of heat and surface heterogeneity, respectively. These findings provide a foundation for the design and optimization of adsorption processes for the removal of metal ions from aqueous solutions. The inhibitor molecules attached to the metallic substrate may be subject to physical or chemical adsorption. The computed slope and intercept values for the Langmuir isotherm were $0.87165 \pm 0.0509$ and $0.13729 \pm 0.02587$, respectively, suggesting that the Langmuir adsorption isotherms matched the data quite well, according to the regression coefficient (R2) for OHMHI of 0.99325. Figure 7 represents the adsorption isotherm plot of Langmuir between $C_{inh}/\theta$ and $C_{inh}$. Based on Equation (14) the adsorption parameters can be determined:

$$C_{inh}/\theta = (K_{ads})^{-1} + C \tag{14}$$

where $C_{inh}$ is the concentration of OHMHI, $\theta$ is the surface area, and $K_{ads}$ is the equilibrium constant.

The adsorption free energy $\Delta G^o_{ads}$ and $K_{ads}$ were evaluated based on the plot between $C/\theta$ and $C$. Equation (15) was used to calculate the adsorption parameters $\Delta G^o_{ads}$ and $K_{ads}$.

$$\Delta G^o_{ads} = -RT \, ln(55.5 K_{ads}) \tag{15}$$

The expression involves the molar concentration of water, which is represented by the value 55.5. This also includes the universal gas constant, denoted by $R$, and the absolute temperature, denoted by $T$.

The chemisorption mechanism is determined by a $\Delta G^o_{ads}$ value of approximately $-40$ kJ·mol$^{-1}$, whereas a $\Delta G^o_{ads}$ value of approximately $-20$ kJ·mol$^{-1}$ refers to a physisorption mechanism [53,54]. A value of $\Delta G^o_{ads}$ for OHMHI of $-38.26$ kJ·mol$^{-1}$ indicates both mechanisms, chemisorption and physisorption. This means that the adsorption of OHMHI on the surface is a combination of both chemical and physical interactions. The negative value of $\Delta G^o_{ads}$ suggests that the adsorption is exothermic and spontaneous. The magnitude of the $\Delta G^o_{ads}$ value indicates that the adsorption process is primarily dominated by chemisorption. This means that the OHMHI molecules are strongly bound to the surface through chemical bonds, resulting in a stable and irreversible adsorption. However, the

proximity of the value to $-20\ kJ\cdot mol^{-1}$ suggests that some level of physical interaction also contributes to the adsorption process. This is typical in many real-world systems where a combination of both chemical and physical interactions occurs.

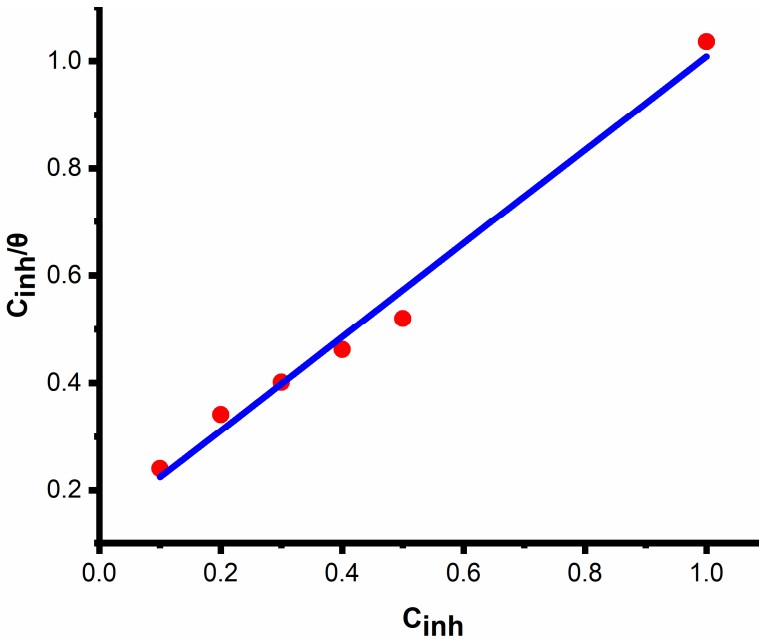

**Figure 7.** Langmuir isotherm for metallic coupon in inhibited acidic solution.

*4.2. Electrochemical Measurements*

4.2.1. PDP

Potentiodynamic polarization measurement curves for a metallic substrate in 1.0 M HCl solution evaluated in different uninhibited and inhibited OHMHI inhibitor concentrations is exhibited in Figure 8. The potentiodynamic polarization measurement curves show the relationship between the applied potential and the corresponding current density in the HCl solution. The curves represent the corrosion behavior of the metallic substrate in the presence of different concentrations of the OHMHI inhibitor. In the uninhibited solution, the corrosion rate of the metallic substrate is high and the potential drops rapidly, indicating a high corrosion rate. However, with the addition of the OHMHI inhibitor, the corrosion rate decreases and the potential drops at a slower rate. This indicates that the inhibitor is effectively reducing the corrosion rate of the metallic substrate. As the concentration of the OHMHI inhibitor increases, the corrosion rate further decreases and the potential drops at an even slower rate. This suggests that higher concentrations of the inhibitor provide better protection to the metallic substrate. The curves also show that the inhibitor has a protective effect on the metallic substrate as the potential remains close to the corrosion potential, indicating a low corrosion rate. The protective effect of the OHMHI inhibitor can be attributed to the formation of a protective film on the surface of the metallic substrate that prevents the corrosion reaction from proceeding. Overall, the potentiodynamic polarization measurement curves provide important information about the effectiveness of the OHMHI inhibitor in reducing the corrosion rate of the metallic substrate in HCl solution.

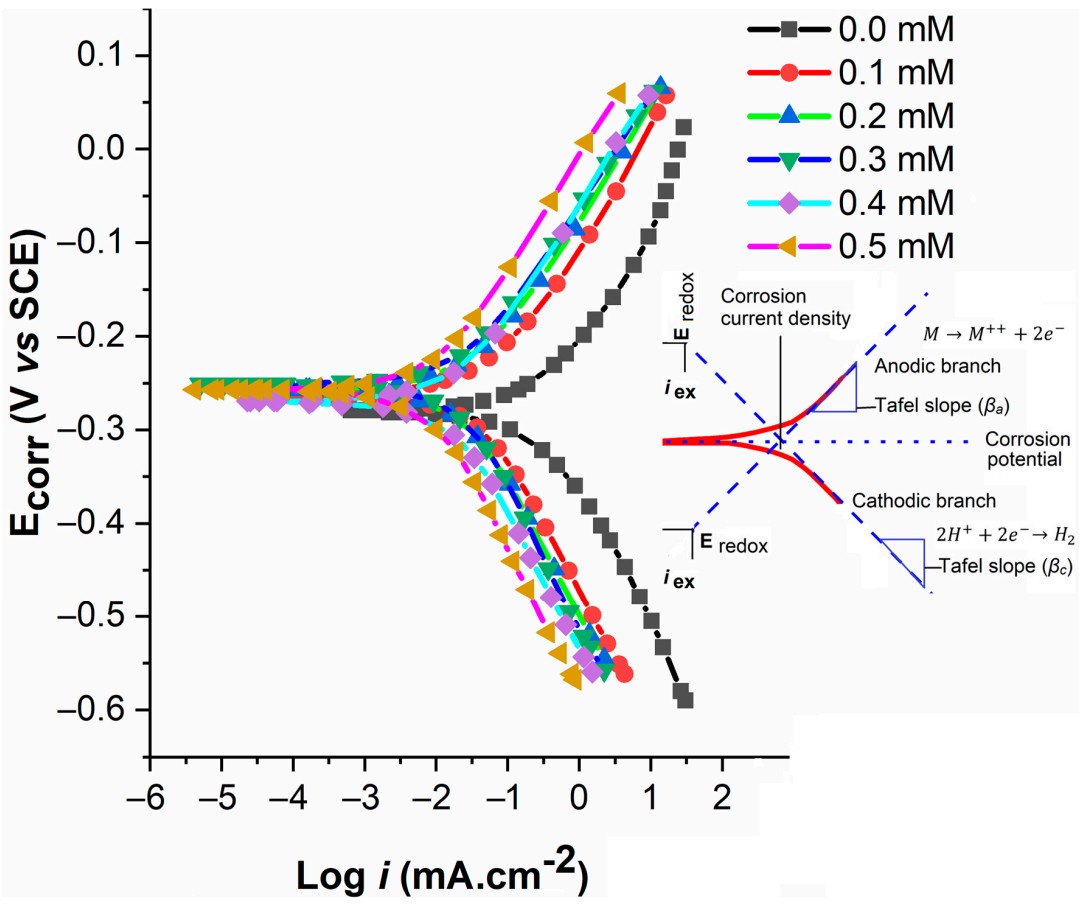

**Figure 8.** Potentiodynamic polarization measurements curves of a metallic substrate in uninhibited and inhibited 1 M corrosive solution at 303 K.

The hydrogen evolution reaction and iron dissolution reaction are the two processes that regulate the rate at which metallic substrate corrodes in a corrosive environment. The steps for anodic breakdown of iron, according to this mechanism, are shown according to Equations (16)–(19) [55].

$$Fe + Cl^- \leftrightarrow \left(FeCl^-\right)_{ads} \tag{16}$$

$$\left(FeCl^-\right)_{ads} \leftrightarrow (FeCl)_{ads} + e^- \tag{17}$$

$$(FeCl)_{ads} \rightarrow \left(FeCl^+\right)_{ads} + e^- \tag{18}$$

$$\left(FeCl^+\right)_{ads} \rightarrow Fe^{++} + Cl^- \tag{19}$$

Equations (20)–(22) can be used to represent the cathodic hydrogen evolution mechanism.

$$(Fe) + H^+ \leftrightarrow \left(FeH^+\right)_{ads} \tag{20}$$

$$\left(FeH^+\right)_{ads} + e^- \rightarrow (FeH)_{ads} \tag{21}$$

$$(FeH)_{ads} + H^+ + e^- \rightarrow Fe + H_2 \tag{22}$$

The cathode reaction becomes significant when the corrosion potential shifts towards the negative, indicating a reduction in the anode reaction. This suggests that the existence of the inhibitors generates a passively protective barrier that inhibits anodic dissolution.

Figure 8 illustrates how the presence of OHMHI lowers the cathodic and anodic currents' corrosive current capabilities. The concentration of an inhibitor affects the rate at which mild steel corrodes. The effectiveness of inhibition is greatly affected by changes in the inhibitor concentration. The anode current densities appear to change somewhat with OHMHI concentration; however, the cathode current densities are significantly impacted. The results in Figure 8 clearly support the finding that, as OHMHI concentration increases, the corrosion potential shifts to a negative value, indicating that OHMHI has a cathodic majority. According to Table 2, the values of the cathodic (c) and anodic (a) Tafel parameters vary slightly when OHMHI is added, suggesting that OHMHI has no effect on the mild steel corrosion process.

**Table 2.** The polarization parameters of metallic substrates in uninhibited and inhibited 1 M HCl solutions at 303 K.

| $C_{inh}$ (mM) | $E_{corr}$ (mV vs. SCE) | $\beta_a$ (mV/dec) | $-\beta_c$ (mV/dec) | $i$ ($\mu$A/cm$^2$) | $IE$ (%) |
|---|---|---|---|---|---|
| 0.0 | −435 | 118.4 | 144.6 | 590 | 58.3 |
| 0.1 | −424 | 92.8 | 117.5 | 140 | 69.5 |
| 0.2 | −422 | 71.7 | 162.5 | 95 | 79.2 |
| 0.3 | −405 | 66.9 | 183.2 | 60 | 84.7 |
| 0.4 | −425 | 53.1 | 171.8 | 88 | 90.2 |
| 0.5 | −410 | 51.8 | 190.4 | 55 | 93.4 |

Furthermore, the observation that the corrosion potentials experience substantial alteration following the inclusion of OHMHI suggests that the inhibition mechanism for this system is not solely due to geometric hindrance, but rather is likely driven by the active site blocking effect [56]. To prevent corrosion, organic inhibitors attach to the metal surface, displacing water molecules and forming a thick barrier coating that blocks the active sites [57].

### 4.2.2. EIS

The determined parameters for electrochemical impedance spectroscopy (EIS) can be found in Table 3, while Figure 9a displays the Nyquist plots for mild steel in inhibited and uninhibited solutions. The Nyquist plots exhibit a single depressed capacitive semicircle in the uninhibited solution, but after OHMHI was added, the impedance modulus of the system reached its maximum value, indicating that the concentration of the inhibitor OHMHI plays a crucial role in the corrosion process. In order to analyze the results of the experiments, we utilized a comparable circuit model illustrated in Figure 9b [58]. The model is composed of the solution resistance (Rs), interfacial corrosion process charge transfer resistance (Rct), and constant phase angle element (CPE). The inhibition efficiency was determined using Equation 5, which employs the charge transfer resistance.

**Table 3.** The EIS parameters for metallic substrates in uninhibited and inhibited 1 M HCl solutions at 303 K.

| $C$(mM) | $R_s$(ohm·cm$^2$) | $R_{ct}$(ohm·cm$^2$) | $n$ | $CPE_{dl}$ $Y_o$ (S·s$^{-n}$·cm$^{-2}$) | $C_{dl}$ | $IE$% |
|---|---|---|---|---|---|---|
| 0.0 | 0.311 | 291 | 0.89 | 241 | 105 | - |
| 0.1 | 0.416 | 420 | 0.87 | 190 | 69 | 60.8 |
| 0.2 | 0.383 | 388 | 0.78 | 118 | 59 | 69.8 |
| 0.3 | 0.512 | 476 | 0.88 | 98 | 50 | 77.4 |
| 0.4 | 0.595 | 541 | 0.77 | 80 | 41 | 87.3 |
| 0.5 | 0.653 | 636 | 0.89 | 58 | 31 | 94.3 |

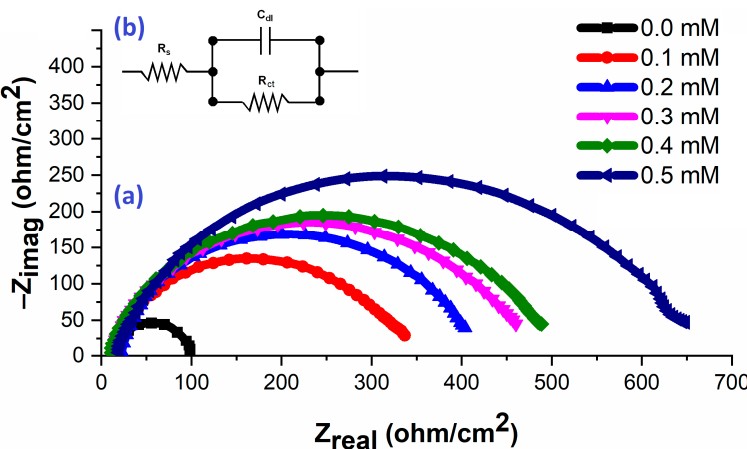

**Figure 9.** (**a**) mild steel Nyquist plots in uninhibited and inhibited corrosive solution. (**b**) Nyquist plots are fitted using an equivalent circuit model.

The EIS findings indicate that the charge transfer resistance value increased as the concentration of OHMHI increased, and the highest inhibition efficiency of 94.3% was achieved at the optimum inhibitor concentration of 0.5 mM. The capacitive loops in the Nyquist plots were not ideal semicircles due to the roughness and non-homogeneity of the metallic substrate causing frequency dispersion [59]. When the OHMHI molecules adsorbed onto the metallic substrate, the $C_{dl}$ decreased, and the double layer thickness increased [60]. We used Equation (23) to determine the value of $C_{dl}$, where $\omega\_max$ is the frequency corresponding to the maximum of -Zimm in the depressed semicircle. This specific frequency value represents the frequency at which the double layer capacitance is most affected by the presence of the OHMHI inhibitor, and it provides a measure of the strength of the inhibitor–substrate interaction. The equation for $C_{dl}$ in terms of $\omega\_max$ is given by:

$$C_{dl} = Y_0(\omega_{max})^{n-1} \tag{23}$$

where $Y_0$ is the coefficient of the constant phase angle element, and $n$ is the exponent of the constant phase angle element. By using this equation to extract the value of $C_{dl}$ from the EIS data, we are able to gain insights into the nature of the inhibitor–substrate interaction and the effectiveness of OHMHI as a corrosion inhibitor.

Equation (24) shows a correlation between the capacitance of the double layer ($C_{dl}$) and the thickness ($d$) of the protective layer.

$$C_{dl} = (\varepsilon\varepsilon_o A)/d \tag{24}$$

where $A$ as the surface area and $\varepsilon_o$ and $\varepsilon$ are the free space permittivity and dielectric constant, respectively.

The Bode plots shown in Figure 10 were obtained using OHMHI. The results indicate that the presence of an adsorbed inhibitor layer increases the capacitive response of the interface, as evidenced by the increased peak heights in the Bode plots [61]. This response is likely due to the formation of an electrochemical double layer with a capacitance. The $C_{dl}$ values observed in the presence of the inhibitor were lower than those obtained without inhibitor, which is consistent with findings reported by other researchers [61–63]. The decrease in $C_{dl}$ values may be due to a reduction in local dielectric or an increase in the thickness of the double layer, both of which are attributed to the adsorbed protective film of the inhibitor molecules as previously suggested [63].

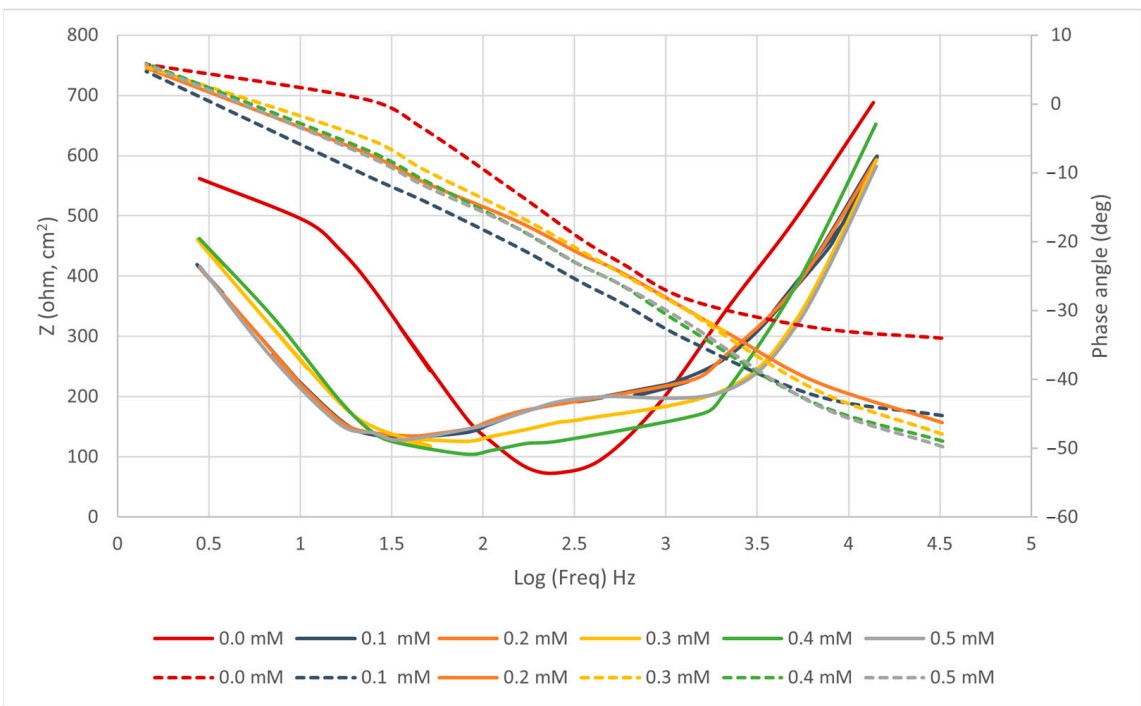

**Figure 10.** Bode plot for mild steel corrosion in 1 M HCl containing different concentrations of OHMHI.

The decrease in $C_{dl}$ values is of significant importance as it reflects a reduction in charge transfer resistance, which in turn results in an increase in the corrosion inhibition efficiency of the inhibitor. The increased capacitance of the interface may be attributed to the formation of a barrier film on the metal surface, which hinders the diffusion of corrosive species towards the metal surface. This barrier film may also act as a physical barrier to protect the metal surface from further corrosion. Overall, the results of this study suggest that the adsorbed inhibitor layer has a beneficial effect on the corrosion inhibition efficiency of the inhibitor. The increased capacitive response and decreased charge transfer resistance observed in the presence of the inhibitor are promising indicators of its effectiveness in inhibiting the corrosion of the metal surface. Further research is needed to fully understand the mechanism of corrosion inhibition and to optimize the use of these inhibitors in industrial applications.

*4.3. DFT*

A number of kinetic parameters can be theoretically derived from Gaussian records depending on the optimized structure (as shown in Figure 11) of OHMHI at B3LYP/6-311G(d,p) when using the quantum chemical approach to analyze the structural features of OHMHI molecules [64,65]. To determine the adsorption centers of an inhibitor, Mulliken charges are employed [66]. The data shown in Table 4 demonstrate that the more negatively charged oxygen and nitrogen atoms in OHMHI are the highest preferred sites for this inhibitor's adsorption onto the metallic substrate via donor–acceptor type of interactions. Additionally, the benzene ring in OHMHI molecules encourages the development of the adsorbate–surface coordination bond complexations [67]. The electron-donating location of the molecule can be seen in the highest occupied molecular orbital (HOMO) [68]. According to Figure 11's depiction of the HOMO of OHMHI, the O and N atoms can primarily transfer electrons to the metallic substrate. The potential of the molecule to receive electrons is indicated by the lowest unoccupied molecular orbital (LUMO), in comparison [67]. OHMHI's reception sites appear to be similar based on the shape of its LUMO, as illustrated in Figure 11. The LUMO positions of nitrogen, oxygen, and carbon atoms in OHMHI are particularly reactive. Various quantum chemical parameters, such as HOMO energy

(EHOMO), LUMO energy (ELUMO), energy gap ($\Delta E$), absolute electronegativity ($\chi$), absolute hardness ($\eta$), global softness ($\sigma$), and the number of electrons transferred ($\Delta N$) were computed and are presented in Table 5. The electron-donating capability of a molecule is associated with its EHOMO value. A suitable acceptor molecule with a low-energy, unoccupied molecular orbital can readily accept electrons from a molecule with a higher EHOMO [68]. According to Table 5, the EHOMO value for OHMHI is low ($-5.646$ eV), which is consistent with the findings of the investigation. The value of ELUMO denotes a molecule's capacity to take electrons. The easier it is for ELUMO to accept electrons, the lower its value must be [69]. The change in chemical potential on the total number of atoms is represented by the absolute hardness ($\eta$) [70]. The stability of a compound increases with the value of ($\eta$) as it increases. Global softness ($\sigma$) is the reverse of hardness and a quantitative feature of electron cloud polarization in compounds [71]. The difficulty of a molecule adsorbing to the metallic substrate increases with the energy gap ($\Delta E$) as it becomes less polar [72]. An excellent corrosion inhibitor must therefore have a low value for $\Delta E$ and $\eta$ but a high value for $\sigma$. OHMHI is a highly effective corrosion inhibitor, according to the computed values in Table 5. Additionally, OHMHI has a high $\Delta N$ value, which denotes a stronger ability to exchange electrons. In summary, the acquired testing findings indicated previously and the quantum chemical parameters (ELUMO, $\Delta E$, $\eta$, $\sigma$, and $\Delta N$) of OHMHI molecules are in excellent accordance. As a result, the link between the metallic substrate and the inhibitor is created by the metal's sharing of electrons with the inhibitor (back-donation) [73]. Each and every quantum chemical parameter (EHOMO, ELUMO, $\Delta E$, $\eta$, $\sigma$, and $\Delta N$) for the studied inhibitor compound exhibit good agreement with the observed experimental outcomes.

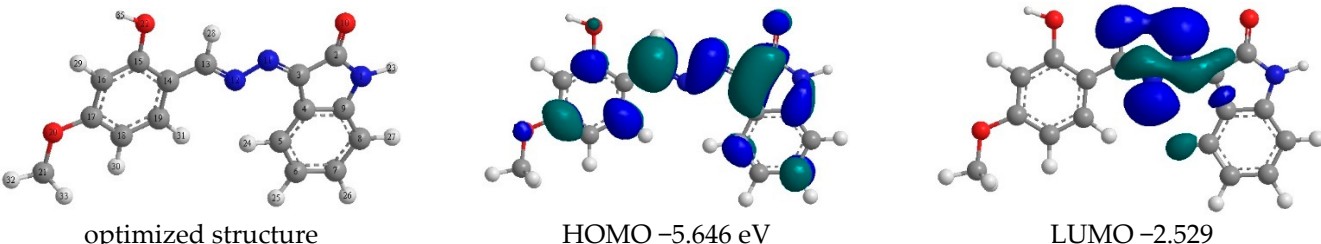

optimized structure      HOMO −5.646 eV      LUMO −2.529

**Figure 11.** OHMHI's structure, as well as its HOMO and LUMO levels, were subjected to optimization.

**Table 4.** The Mulliken charges of OHMHI in gas phase.

| Atoms | Charges | Atoms | Charges | Atoms | Charges | Atoms | Charges |
|---|---|---|---|---|---|---|---|
| N − 1 | 0.353 | C − 7 | −0.139 | C − 13 | −0.229 | C − 19 | −0.154 |
| C − 2 | 0.276 | C − 8 | −0.142 | C − 14 | −0.055 | O − 20 | −0.233 |
| C − 3 | −0.110 | C − 9 | 0.095 | C − 15 | 0.190 | C − 21 | 0.078 |
| C − 4 | −0.075 | O − 10 | −0.928 | C − 16 | −0.185 | O − 22 | −0.277 |
| C − 5 | −0.118 | N − 11 | 0.599 | C − 17 | 0.138 | H − 23 | 0.078 |
| C − 6 | −0.132 | N − 12 | 0.665 | C − 18 | −0.146 | H − 24 | 0.026 |

**Table 5.** The quantum chemical parameters of OHMHI molecules were analyzed in the gas phase using the B3LYP/6-311G(d,p) method.

| *I* | *A* | $E_{HOMO}$ | $E_{LUMO}$ | $\Delta E$ | $\chi$ | $\eta$ | $\sigma$ | $\Delta N$ |
|---|---|---|---|---|---|---|---|---|
| 5.646 eV | 2.529 eV | −5.646 eV | −2.529 eV | −3.117 eV | 4.0875 eV | 1.5585 eV | 0.6416 eV$^{-1}$ | 0.93433 |

*4.4. Surface Analysis*

SEM photographs of the immersed mild steel coupons in uninhibited and hydrochloric acid environments are shown in Figure 12. It is obvious from Figure 12a that the coupon surface was badly damaged in uninhibition solution and exhibits a number of pits and

cracks. The mild steel in inhibited solution as shown in Figure 12b was noticeably improved with fewer cracks and pits as compared with the mild steel surface in uninhibited solution. This enhancement is the result of the mild steel's surface forming a prohibitive film. Owing to the oxygen and nitrogen atoms having unpaired electron pairs in addition to the way in which the pi-electrons of double bonds inhibit active sites and therefore slow the corrosion rate, mild steel surfaces have strong inhibitory efficacy.

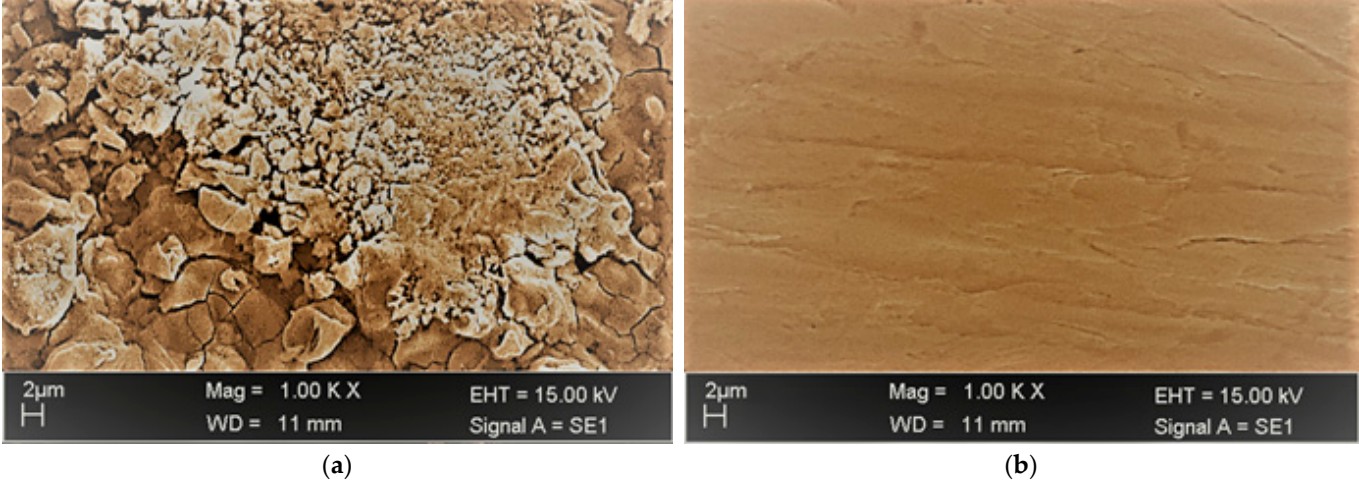

(**a**)  (**b**)

**Figure 12.** (**a**) SEM image of mild steel in uninhibited solution, (**b**) SEM image of mild steel in inhibited solution.

### 4.5. Suggested Mechanism

The corrosion of mild steel in 1 M hydrochloric acid can be prevented by OHMHI through a process known as adsorption. However, it is difficult to understand the mechanism of adsorption as inhibitors can be adsorbed through both physical and chemical interactions. In the case of OHMHI, it is likely that inhibitor molecules attach themselves to the surface of mild steel through van der Waals forces, which is a physical process. This forms a protective layer between the metal and the acid, thus reducing the corrosion rate. Moreover, OHMHI can also react chemically with the corrosion products present on the metal surface, further reducing the corrosion rate. For instance, OHMHI may react with the iron oxides formed on the steel surface to form a protective film, which hinders further corrosion. Overall, the adsorption process of OHMHI on the surface of mild steel in 1 M hydrochloric acid is complex, involving both physical and chemical interactions, which effectively inhibits corrosion. Figure 13 illustrates several methods that can be used to describe the adsorption process [74].

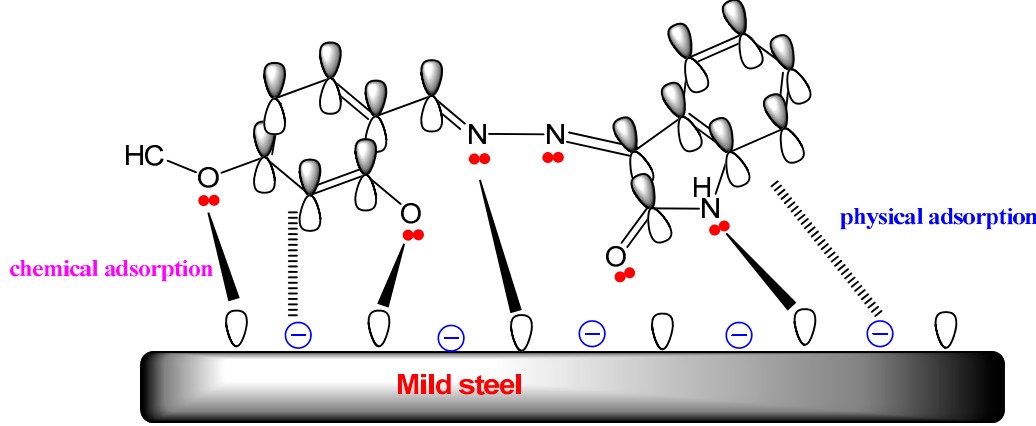

**Figure 13.** The suggested inhibition mechanism of OHMHI as corrosion inhibitor.

- Protonated molecules (ions) of the OHMHI are present in corrosive media. Through electrostatic contact, these ions are deposited on the mild steel interface where $Cl^-$ had earlier been adsorbed (physical adsorption).
- Through the chemical adsorption process, OHMHI can be adsorbed on the metallic surface by donating lone pairs of electrons from nitrogen and oxygen atoms to the vacant orbital of iron atoms.
- Pi-electrons from benzene ring and the unoccupied d-orbital of iron atoms interact as donors and acceptors [75].

## 5. Conclusions

In conclusion, the study of OHMHI as a corrosion inhibitor for mild steel in 1 M hydrochloric acid solution has demonstrated its excellent protection performance. The inhibition effectiveness of Isatin Schiff base was found to be 96.1% at the optimum concentration and 303 K, as measured by weight loss. The effects of concentration, immersion time, and temperature were also examined, revealing OHMHI's mixed-type inhibition behavior, with a primary shift towards the cathodic direction. Polarization research suggests that OHMHI primarily acts as a cathodic inhibitor. Additionally, EIS findings show that the inhibitor molecules adhered to the mild steel surface, leading to an increase in charge-transfer resistance ($R_{ct}$) and a decrease in $C_{dl}$. The Langmuir adsorption isotherm was found to govern the adsorption of inhibitor molecules on the mild steel surface, with SEM images revealing that the inhibitor molecules shielded the mild steel surface from corrosive ions. Finally, quantum chemical studies using density functional theory (DFT) agreed well with the experimental data. Overall, the results highlight the potential of OHMHI as an effective inhibitor for protecting mild steel against corrosion in acidic environments.

## 6. Compared with Other Published Materials, What Does Ours Bring to the Field?

The use of Isatin–Schiff base derivatives as corrosion inhibitors for mild steel in HCl solution has been extensively studied in recent years. Compared with other published corrosion inhibitors, the Isatin–Schiff base derivatives have several advantages:

High inhibition efficiency: Isatin–Schiff base derivatives have been found to exhibit high inhibition efficiency against mild steel corrosion in HCl solution. In some studies, inhibition efficiencies as high as 98% have been reported.

Low toxicity: Isatin–Schiff base derivatives are generally considered to be safe and have low toxicity. This makes them more attractive as corrosion inhibitors for industrial applications.

Cost-effective: Isatin–Schiff base derivatives are relatively inexpensive and can be easily synthesized in the laboratory. This makes them a cost-effective alternative to other corrosion inhibitors.

Environmentally friendly: Isatin–Schiff base derivatives have also been found to be environmentally friendly and biodegradable, which is a crucial aspect for sustainable development.

Overall, the use of Isatin–Schiff base derivatives as corrosion inhibitors for mild steel in HCl solution brings several advantages to the field, including high inhibition efficiency, low toxicity, cost-effectiveness, and environmental friendliness. These properties make them a promising candidate for industrial applications.

**Author Contributions:** Conceptualization, W.N.R.W.I.; methodology, N.B.; software, N.B.; validation, W.M.N.W.N.; formal analysis, W.K.A.-A.; investigation, A.A.A.-A.; resources, W.N.R.W.I.; data curation, N.B.; writing—original draft preparation, A.A.A.-A.; writing—review and editing, A.A.A.-A.; visualization, W.K.A.-A.; supervision, W.N.R.W.I.; project administration, W.M.N.W.N.; funding acquisition, W.N.R.W.I. All authors have read and agreed to the published version of the manuscript.

**Funding:** Universiti Kebangsaan Malaysia provided funding for a portion of the study under the following code: GUP-2020-012.

**Institutional Review Board Statement:** Not applicable.

**Informed Consent Statement:** Not applicable.

**Data Availability Statement:** Not applicable.

**Acknowledgments:** The support provided by the Universiti Kebangsaan Malaysia (UKM) is acknowledged by the authors. We would like to acknowledge also the joint UKM-UMT-RSU-AISSMS (1+3) research collaboration in the field of corrosion.

**Conflicts of Interest:** The authors declare no conflict of interest.

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
