# Peer review of "Exploring the Effectiveness of Isatin–Schiff Base as an Environmentally Friendly Corrosion Inhibitor for Mild Steel in Hydrochloric Acid"

_lubricants, doi:10.3390/lubricants11050211_

Round 1
Reviewer 1 Report
Al-Amiery et al. examine the Isatin Schiff base as green corrosion inhibitor for mild steel in hydrochloric acid environment. The article is written nicely, explaining all the aspects related to the study. It will help researchers who are working in the same field. I recommend its acceptance after a major revision followed by the editorial correction.
1. Although the abstract is long, but it should contain key findings of the work, please add it.
2. Add more literature in the introduction section.
3. Compared to other published materials, what does it bring to the field?
4. There should be a space (gap) between the numerical values and the units.
5. What makes this work novel?
6. The author should contrast this recent study with previously published authoritative data.
7. The social and economic effects of corrosion might be added by the authors.
8. Table 2, “-Ecorr” should be “Ecorr”.
9. Data present in Table 2, does not match with the Figure 8 (Ecorr values).
10. Data present in Table 3 does not match with the Figure 9, (Rct values) check and correct it.
11. Fitted curves are missing from figure 9.
12. The conclusion can be improved.
13. Please exchange old papers with newly published papers.
14. Please follow one homogeneous format style of referencing. Different styles of referencing can be seen.
15. Please check the entire manuscript and remove grammatical errors.
16. Plagiarism can be seen (over 30%) please reduce it by <20%.
Author Response
Dear Reviewer,
Thank you very much for taking the time to review our manuscript entitled "Isatin Schiff base as Green Corrosion Inhibitor for Mild Steel in Hydrochloric Acid Environment" submitted to Lubricants. We greatly appreciate your valuable feedback, which has helped us to improve the quality of our research.
We have carefully considered all of your comments and have made the necessary revisions to the manuscript accordingly. In response to your specific points.
Please see the response letter and the revised manuscript.
Thank you
Best regards

Reviewer 2 Report
The paper is well-written and has a robust scientific basis. Some suggestions from my side are here reported:
* Introduction:
"In comparison to inhibitors containing only one of the two elements, those combinations of both nitrogen and sulfur are incredibly effective at preventing corrosion": I suggest adding some examples to make the concept clearer. Then, consider that sulfur is not present in the investigated molecule: a discussion about the effects of different hetero atoms must be added.
"The results showed that OHMHI was effective in reducing the corrosion rate of mild steel in hydrochloric acid solution, with the highest inhibition efficiency of 96.1% recorded at a concentration of 5 mM. Surface analysis indicated that OHMHI adsorbed on the mild steel surface and formed a protective film that prevented the reaction between the metal and the corrosive solution. In conclusion, OHMHI can be used as an effective corrosion inhibitor for mild steel in hydrochloric acid solution.": I suggest removing all these sentences because the introduction is not abstract.
I suggest adding to the introduction some sentences explaining which is the innovation of this paper: the molecule you used? the experimental conditions of corrosion? the experimental techniques?
* Materials and methods:
"Coupons were cleaned based on the standard technique G1-03/ASTM [17]": I suggest adding a summary of the surface preparation
"were abraded with a series of emery paper": I suggest being more detailed and giving the numbers of the abrasive papers you used.
* Results and discussion:
"The effect of different concentrations of corrosion inhibitors on mild steel in 1 M HCl solution will vary. Generally, higher concentrations of corrosion inhibitors will result in better protection against corrosion, but the optimal concentration may depend on the specific inhibitor and other variables, such as temperature and the presence of other ions in the solution. Additionally, some inhibitors may become less effective at high concentrations or even become corrosive themselves. It's best to refer to literature and studies specific to the chosen inhibitor for more accurate information. In the losing mass test for metallic coupon in uninhibited and inhibited corrosive solution, it was found that the presence of OHMHI (a corrosion inhibitor) protected the metallic surface from the corrosive solution.": I suggest following a standard order: the new results of this research must be firstly presented, then compared internally and with the literature, and lastly the discussion must be performed. In this sentence, there are conclusions (phrase 5) before the results descriptions and the first 4 phrases have no references and are generic.
"Since the OHMHI has a large molecular structure and contains ..... to the strong interaction of OHMHI with the metal surface, resulting in an effective corrosion inhibitor [25].": This long part is not needed at this point and it is not correlated to the reported results. I suggest being much more strictly focused on the obtained results: a comparison with the literature data about similar molecules concentrations and corrosion rates can be added.
"Moreover, the inhibitory efficacy of OHMHI is enhanced in the acidic environment due to the coordination interactions between the inhibitor molecules and the iron atoms on the mild steel substrate": Actually, you are not reporting here data about different pH and this sentence is not experimentally justified. I suggest removing it.
"The decrease in inhibition efficiency at higher temperatures can also be attributed to the thermodynamic instability of OHMHI at elevated temperatures": I suggest reporting a reference and a range of temperatures of stability.
"OHMHI performed best when the temperature was normal": what does "normal" mean in this context? is it room temperature?
"Metallic substrate immersing process is endothermic": I suggest specifying which process you are referring to: is it corrosion?
"the presence of the inhibitor has reduced the activation energy required for the corrosion process to occur": I suggest avoiding ambiguous sentences: if the activation energy for corrosion is lower, it means that corrosion is thermodynamically advantaged. On the other side, if the delta G between the products and reagent is lower it means that the reaction of corrosion is disadvantaged.
Can you add a paragraph to the materials and methods section concerning how the adsorption isotherms were obtained?
"The inhibitor molecules adsorbed on the metallic substrate would be physical and/or chemical adsorption": I suggest rephrasing
"where Cl had earlier been adsorbed": is it the ion Cl-?
"by donating lone electron": check for this phrase
Author Response
Dear Reviewer,
Thank you for your valuable feedback on our manuscript. We appreciate your time and effort in reviewing our work.
We are happy to inform you that we have carefully considered all your comments and have made the necessary corrections to address the issues raised. We believe that these changes have significantly improved the quality of our manuscript and have strengthened the overall argument.
Once again, thank you for your thoughtful feedback, which has helped us to enhance the quality of our work. We hope that the revised manuscript meets your expectations, and we look forward to hearing back from you soon.
Best regards,
Ahmed Al-Amiery

Round 2
Reviewer 1 Report
Data present in Table 2 and Table 3 do not sink with the corresponding figures—please revise the table data. follow this paper:
https://doi.org/10.1016/j.molliq.2022.121168
Minor grammatical errors can be seen. carefully revise the entire manuscript.
Author Response
Dear Reviewer,
Thank you for taking the time to review our manuscript and providing your valuable feedback. We appreciate your effort in reviewing our work and your suggestions for improving the manuscript.
We have carefully reviewed your comments and suggestions and have taken them into consideration for the revision of the manuscript. We acknowledge the mismatch between the data presented in Table 2 and Table 3 and the corresponding figures, and we will revise the tables accordingly to ensure that the data is accurately presented.
Furthermore, we apologize for any minor grammatical errors that may have been present in the manuscript. We will thoroughly review the entire manuscript to correct any grammatical errors and improve the overall language and readability of the manuscript.
Once again, we appreciate your feedback and valuable suggestions, and we hope that the revised manuscript will meet your expectations.
Thank you for your time and attention.
Sincerely,
Ahmed Al-Amiery
Reviewer 2 Report
I congratulate the authors for the effective revision of the paper.
I still suggest removing the "normal temperature" phrase and using the number ("at the temperature of 303K").
Some experimental details can be added to paragraph "3.4. Adsorption isotherms": the type of scale you used, the area of the samples ....
Author Response
Dear Reviewer,
Thank you for taking the time to review our paper and for your positive comments on the effective revision of the manuscript.
Regarding your suggestion to remove the phrase "normal temperature", we agree with your point and have updated the manuscript to use the specific temperature of 303K throughout the text.
We also appreciate your suggestion to add more experimental details to paragraph 3.4 on the adsorption isotherms. We have now included information on the type of scale used, as well as the surface area of the samples, to provide readers with a more comprehensive understanding of our methodology.
Thank you again for your valuable feedback, which has helped us to improve the quality of our paper. We hope that these revisions meet your expectations, and we look forward to hearing from you soon.
Best regards,
Ahmed Al-Amiery